# Implementation of MALDI Mass Spectrometry Imaging in Cancer Proteomics Research: Applications and Challenges

**DOI:** 10.3390/jpm10020054

**Published:** 2020-06-22

**Authors:** Eline Berghmans, Kurt Boonen, Evelyne Maes, Inge Mertens, Patrick Pauwels, Geert Baggerman

**Affiliations:** 1Centre for Proteomics, University of Antwerp, Groenenborgerlaan 171, B-2020 Antwerpen, Belgium; eline.berghmans@vito.be (E.B.); kurt.boonen@vito.be (K.B.); inge.mertens@vito.be (I.M.); 2Health Unit, VITO, Boeretang 200, 2400 Mol, Belgium; 3Food & Bio-Based Products, AgResearch Ltd., Lincoln 7674, New Zealand; evelyne.maes@agresearch.co.nz; 4Department of Pathology, Antwerp University Hospital, Wilrijkstraat 10, 2650 Edegem, Belgium; patrick.pauwels@uza.be; 5Center for Oncological Research, University of Antwerp, Universiteitsplein 1, 2610 Wilrijk, Belgium

**Keywords:** MALDI mass spectrometry imaging, proteomics, proteomic profiling, cancer research, personalized medicine

## Abstract

Studying the proteome–the entire set of proteins in cells, tissues, organs and body fluids—is of great relevance in cancer research, as differential forms of proteins are expressed in response to specific intrinsic and extrinsic signals. Discovering protein signatures/pathways responsible for cancer transformation may lead to a better understanding of tumor biology and to a more effective diagnosis, prognosis, recurrence and response to therapy. Moreover, proteins can act as a biomarker or potential drug targets. Hence, it is of major importance to implement proteomic, particularly mass spectrometric, approaches in cancer research, to provide new crucial insights into tumor biology. Recently, mass spectrometry imaging (MSI) approaches were implemented in cancer research, to provide individual molecular characteristics of each individual tumor while retaining molecular spatial distribution, essential in the context of personalized disease management and medicine.

## 1. Proteins, the Core Elements of Our Cells

Through the mapping of all the human genes in the Human Genome Project (HGP), important insights into the genomics of cancers were made, as many mutated genes are responsible for tumor development/growth. These genetic insights have helped to develop new diagnostic and targeted therapeutic tools [1]; as an example, mutation(s) in the epidermal growth factor receptor (EGFR) is frequently observed in non-small cell lung cancer (NSCLC) patients. When mutated, EGFR is continuously activated, which leads to uncontrolled lung tumor growth. Specific targeted therapies (e.g., erlotinib and afatinib) have been developed to block the activity of EGFR, but resistance to these therapies nearly always occurs in these patients [2]. This indicates that ‘post-genomic’ research is essential to fully understand cancer mechanisms for development and progression.

The balanced conversion of DNA to RNA to proteins is crucial for a normal physiological state; the genes are the so called controllers of the cell, however, these controllers do not accurately predict the expression level of the protein or whether the protein will be stable and functional. Thereby, a single gene can give rise to many proteoforms, due to single amino acid polymorphisms, alternative splicing (isoforms) and post-translational modifications (PTMs). These proteoforms are molecularly distinct proteins and can modulate different cellular functions. Proteins are therefore the effectors involved in cellular processes and signaling and malfunctioning or misfolding of proteins have been associated with various diseases, for example cancer and Alzheimer’s disease [3]. Studying proteins in the context of cancer is of major relevance to understand the underlying biology at the molecular level; implementing proteomic approaches in cancer research can provide new crucial insights into tumor biology by analyzing protein expression levels and modifications resulting in different functioning. Moreover, proteomics can reveal new pathways involved in cancer progression and may lead to new therapeutic approaches [4].

Proteins can be analyzed by mass spectrometry (MS); ionization of the proteins leads to measurement of the mass-to-charge (*m*/*z*) ratio, with the great benefit that the analysis is unbiased and a priori knowledge of the proteins is thus not necessary. Using mass spectrometry, not only the protein content in different types of samples such as blood, serum, urine and tissues can be identified, but even the different proteoforms within each sample can be detected with top-down proteomics. In addition, mass spectrometry is a (semi-)quantitative technique, which allows comparison of protein abundances between different conditions [4,5]. Furthermore, protein interactions can be studied using mass spectrometric approaches and even PTMs can be identified and mapped [3]. This is of great interest as PTMs such as phosphorylation, glycosylation, acetylation and proteolytic processing are common events and can have tremendous functional consequences, resulting in modification of the characteristics of cells [6]. In conclusion, proteins are the effectors of the cell and have a greater information load than the gene. As a consequence, proteomic profiling of cancers, called cancer proteomics or oncoproteomics, can provide crucial insights into tumor biology beyond what can be learned from genetic analysis [5]. Therefore, mapping the human cancer proteome to unravel and better understand tumor biology is of high importance, as is illustrated by the human cancer proteome project (cancer-HPP) from the Human Proteome Organization (HUPO). The aim of which is to help develop improved diagnostics and new leads for treatment of cancer [7].

## 2. Proteomic Profiling of Cancer

When comparing samples of different biological conditions, e.g., tumor versus healthy state, quantitative proteomics generates lists of hundreds to thousands of proteins that are differentially expressed, indicating abnormal presence, absence or changes in expression of certain proteins, change in PTMs, as a consequence of ongoing physiological or pathological events [8]. These important biological indicators, so-called biomarkers, represent powerful tools for monitoring cancer progression and in general for the understanding of cancer biology, a highly dynamic process, which may improve healthcare [3]. Such measurable analytes are important as they can be used to differentiate biological conditions that are benign or malignant [9]. For protein biomarker(s), either a single entity or a panel of protein markers (protein signatures or proteomic profiles), the presence or change in abundance of specific analytes indicates certain features of a normal physiological or disease state [9]. Detection and identification of protein biomarkers can aid clinicians and scientists in cancer diagnosis, status, progression and assessment of efficacy and toxicity of therapeutic agents (i.e., pharmacological response) in the context of personalized medicine, for which a schematic overview is represented in Figure 1. Due to their high specificity and sensitivity, mass spectrometry is a powerful tool to obtain such protein signatures or proteomic profiles [8,9].

Biological fluids, f.e. urine, blood, serum and plasma, are the most commonly used specimens for protein biomarker research, as sufficient amounts of these types of specimens can be easily obtained without invasive collection, although blood collection can be considered as mildly invasive sample collection. Thereby, they frequently contain proteins secreted from the tumor. As the tumor surrounding stroma also secretes proteins into the circulation as a response to nearby tumor growth, protein profiling can be a reflection of the body’s immune response activated by cancer [5]. As an example, MS-based protein patterns in serum from colorectal cancer (CRC) patients and healthy individuals were compared to classify differentiating proteins. Coupled with sophisticated bioinformatics tools, Yu et al. [10] found four potential protein biomarkers that were highly expressed in CRC patients and low in healthy individuals. The resulting protein fingerprint has the potential to diagnose cancer in a patient. The aim is to allow earlier detection of this type of cancer, which can result in a better prognosis: higher chances of recovery by conventional therapies and thus decreased disease-related mortality [10]. In another study, quantitative proteomics was used to analyze blood samples from pancreatic ductal adenocarcinoma (PDAC) patients in search of prospective biomarker discovery for chemotherapy outcome and PDAC patient survival time [11]. Peng et al. [11] identified a composite biomarker panel of four specific proteins differentially expressed in plasma from PDAC patients. This panel can discern positive responding patients with longer overall survival from those who do not respond well to the administered chemotherapy, resulting in shorter survival time. Identification of such therapy predictive biomarkers is important to administer therapy only to those patients responding to the therapy to minimize harm and cost, while maximizing patient benefit. Additionally, the dual value of proteomics can also facilitate future drug development; Zhang et al. [12] reported 82 differentially expressed proteins in the context of renal cell carcinoma (RCC). Overexpression of one of the upregulated proteins, progesterone receptor membrane component 1 (PGRMC1), was significantly associated with renal cancer cell proliferation, while silencing of PGRMC1 resulted in the opposite phenomenon, demonstrating PGRMC1 as a novel potential therapeutic target for RCC.

As described above, although liquid biopsies are easily obtained, protein secretion into the circulation may be influenced by many factors, including biological heterogeneity or other complications due to malignancies or inflammatory responses [11]. As a consequence, due to dilution of the potential specific protein markers in the circulation, most of the identified markers in these types of samples are limited to the highly abundant proteins. Evidently, lower abundant proteins have high potential interest and may be more specific than the highly abundant proteins for certain conditions. Investigating low abundant proteins is often not possible from liquid biopsies as, in contrast to RNA or DNA, no amplification reaction can be performed on proteins. Therefore, to study low abundant proteins, one has to resort to studying the tumor tissue itself and use it to unravel cancer proteomic signatures [9]. Using matrix-assisted laser desorption/ionization time of flight (MALDI-TOF) mass spectrometry (MS) in homogenized lung tumor tissues mixed with a suitable matrix for ionization and thus detection of proteomic content, Yanagisawa et al. [13] were the first to detect proteomic patterns to classify lung tumor versus normal lung tissue with a combined use of class-prediction models. Furthermore, the model can also distinguish primary non-small cell lung cancer from cancer metastasis to the lung and additionally, histologic classification of the three different NSCLC types adenocarcinoma, squamous cell carcinoma and large-cell carcinoma can be made [13]. The right diagnosis and site of origin of tumors is crucial for selecting the right treatment. In 2007, the same researchers also constructed a signature for prognosis of NSCLC patients using MALDI-TOF MS [14]. In this study, proteomic patterns from resected NSCLC tissues were compared between patients at high and low risks of relapse. Sophisticated bioinformatic analyses revealed 25 *m*/*z* signals that were associated with a high risk of NSCLC recurrence (within 5 years after resection). These models can be used to avoid overtreatment of patients who are not likely to relapse [14]. Determination of different profiles specific for different stages of lung cancer development, was performed by Rahman et al. [15] with MALDI MS on lung epithelium tissues. Statistical analysis of these proteomic patterns revealed *m*/*z* signals that could discriminate for normal epithelium (differentiation between alveolar and bronchial), low-grade preinvasive lesions, high-grade preinvasive lesions and invasive lung tumors. Further, a subset of this proteomic signature (9 *m*/*z* signals) may facilitate the diagnosis of lung cancer and monitoring high risk individuals for lung cancer, as only a subset (30–50%) of preinvasive lesions progresses to invasive lung tumors, which cannot be predicted by histologic interpretation alone [15]. Another example of proteome analysis was performed on normal and ovarian tissue specimens, where a new predictor of prognosis was established for serous ovarian carcinoma (SOC), the most aggressive form of ovarian cancer [16]. In this study, six proteins were identified that may be involved in development or progression of SOC. For one protein in particular, glia maturation factor β (GMFB), high expression was correlated with lower disease-free and overall survival rates, which makes GMFB suitable as a predictor for SOC patient prognosis and survival [16]. A last example of successful tissue-based proteomic profiling is the discovery of predictive biomarkers to predict the patient’s response to a certain therapy, important for effective cancer treatment. Yang et al. [17] discovered two predictive markers (i.e., FKBP4 and S100A9) to neoadjuvant chemotherapy in breast cancer patients by proteomic and bioinformatic approaches. Pre-treatment needle-biopsy breast cancerous tissues were obtained and patients were divided in drug resistant and drug sensitive (i.e., more than 30% reduction in tumor size after chemotherapy) groups. Quantitative proteomic analysis, validated with immunohistochemical analyses, revealed that overexpression of FKBP4 and low expression of S100A9 were associated with chemotherapy resistance [17]. From all these examples, the benefit of using proteomics as a discovery tool in cancer research has been demonstrated to provide new understandings in pathological states and has led to the identification of protein biomarkers for diagnosis, prediction of both disease progression and a response to certain treatments.

## 3. MALDI Mass Spectrometry Imaging in Cancer Research

Ideally for clinical analyses, specimens requiring minimal invasive techniques (f.e. blood and urine) will be of first choice for analysis. However, tissues are the most relevant biological material for providing new insights into disease mechanisms, as protein concentration can diminish with increased distance from the tumor due to dilution effects [5]. Tissues have the additional advantage that the tumor microenvironment can also be taken into account, containing a high amount of proteins for discovery of new potential biomarkers. Solid tissues are spatially complex samples, and therefore, linking the molecular information to tissue morphology is often crucial for the correct biological interpretation. This can lead to a better understanding of the intra- and intertumoral heterogeneity [18]. Diversity occurs between and within tumors, as the same tumor type can show a patient-specific unique combination of genomic alterations, i.e., intertumoral heterogeneity. Additionally, within a tumor, cells can have different molecular characteristics as a simple consequence of the deficiency of DNA replication, called intratumoral heterogeneity. Both heterogeneities introduce important challenges for cancer treatment and stratification of patient populations who are likely to benefit from specific treatments. However, tumor heterogeneity is an important factor to investigate and better understand cancer development and progression [18].

To this end, MALDI mass spectrometry imaging (MSI) can be a powerful tool to provide molecular information while retaining the spatial distribution of the molecules throughout the tissue. With MALDI MSI, a mass spectrum of each spot of a tissue slice is generated. This forms a map of the biomolecules present in that spot of the tissue. Subsequently, all of the individual recorded mass spectra are merged in one resulting overall average mass spectrum. The measurements are taken in a predefined order on a raster, and this allows to analyze both the distribution and the relative abundances of each biomolecule over the entire tissue section [19]. While this review is limited to proteomic studies, it is worthwhile to note that MALDI MSI can produce reliable images of the spatial distribution from a broad variety of biomolecules, ranging from peptides, to glycans, lipids and even metabolites, known to play important roles in cancer [20,21,22,23]. One such example is a MALDI MSI lipidomic profiling study which revealed 10 lipids, differentially expressed in a metastasizing medulloblastoma compared to a non-metastasizing one. This finding could provide a better understanding of medulloblastoma progression and to the discovery of novel biomarkers for the prevention of metastasis [23]. Metabolite biomarkers have also been identified using MALDI MSI on tumor tissues: an example is given by Lou et al. [22] who discovered that high concentrations of the metabolite inositol cyclic phosphate was associated with poor overall survival in soft tissue sarcomas, while carnitine has been identified as a poor metastasis-free survival metabolic biomarker in soft tissue sarcoma patients [22]. Finally, N-glycan imaging on pancreatic tissues was performed by Powers et al. [21] and revealed four different glycans that could distinguish between normal and tumor pancreatic tissue, in addition to differentiating regions of desmoplasia from the necrotic region [21]. The same study also showed similar N-glycan patterns that could distinguish between prostate stroma and gland.

Recent technical developments have evolved MALDI MSI into a high speed analysis approach with increased resolution, mass resolving power and mass accuracy, enabling high-throughput analyses at a speed that is comparable to routine clinical analysis such as immunohistochemical (IHC) analysis. This makes MALDI MSI an interesting biomedical tool in cancer research, but possibly also for clinical assays, especially since MSI is not restricted to the study of a single protein (like IHC), so with one analysis a systemic overview can be generated allowing a better understanding of heterogenous diseases at the cellular and molecular level [6]. For every *m*/*z* value in the resulting MALDI MSI spectrum, the distribution of this molecule of interest can be visualized within the tissue. With recent software developments, visualization is possible of both individual molecules and groups of molecules. The latter is of interest to perform proteomic profiling on regions of interest on the tissue section itself. In this way, regions-of-interest (ROIs), determined with subsequent histological (f.e. H&E) staining on the same tissue section, can be extracted virtually to generate mass spectrometric profiles according to the specific ROI [24]. An overview is illustrated in Figure 2, where MSI profiles of normal gastric mucosa and gastric carcinoma are shown [25].

Every individual MALDI MSI experiment generates a large amount of mass spectra. Clinical MSI high-throughput experiments can involve analysis of hundreds of tissue samples, making the processing and handling of clinical MSI data rely heavily on computational methods. Available software packages, f.e. SCiLS lab (SCiLS, Bremen, Germany), Cardinal [26] or msIQuant [27], can provide many statistical analyses on terabyte-sized multiple samples for recognition of different patterns, classification of different biological regions, biomarker discovery, etc. [18]. Also important for implementation of MALDI MSI in clinical research is the possibility to combine MSI and classical microscopy images into a single image from both sources to provide more accurate information about the tissue sample [28]. In this way, the coarse spatial resolution images obtained with MSI describe important chemical information and need to be combined with high spatial microscopy images to sharpen the resulting fused image. The non-destruction capability of MALDI MSI makes a co-registration with histological data possible and this allows MSI-microscopy fusion on a single tissue specimen. This fusion process requires extensive modeling as a massive multivariate regression task involving variables derived from both modalities to combine in a single predicted modality, presented in Figure 3. This remarkable combination can reveal insights that are otherwise not easily obtained by either microscopy or MSI alone [28].

A disadvantage of MALDI MSI is the fact that identification of interesting MSI targets straight from the tissue itself is still cumbersome, which hampers the validation of candidate biomarkers (f.e. by immunohistochemical analyses that require the identity of the putative biomarker) and the discovery of biological processes that might underlie the observed disease [29]. Nonetheless, implementation of MSI in clinical research has been aided by the initiation of openly available databases of identifications. Within the MSiMass list database [30], information is provided to assign protein identities to the observed MSI peaks, accompanied with the used method to confirm identification of the observed peaks. MSiMass list is an open community and every MSI researcher can further complete the MSiMass list with recent developments and identifications, to facilitate the integration of MSI in clinical research [30]. Another method to achieve the identification of MSI targets is the implementation of Liquid Extraction Surface Analysis (LESA) procedure. For this methodology, on-tissue trypsin digestion is first performed followed by micro-extraction of the tryptic peptides on a well-controlled area. Micro-extraction includes contact of an organic solvent with the sample surface in terms of discrete droplets to extract molecules in a very small and specific area. The extracted molecules can then undergo additional sample preparation steps before analysis on higher-resolution mass spectrometry instruments for identification. Quanico et al. were able to identify in this way about 1500 proteins in a small area of 670 µm in diameter in a bottom-up proteomics approach [31]. They incorporated this methodology in the MALDI MSI workflow, where they used first classification methods on MSI images to determine regions-of-interest and these ROIs were then subjected to on-tissue digestion and micro-extraction for identification of the observed proteins [31]. More recently, the same research group was able to identify more than 500 proteins in a region as small as 250 µm in diameter [32].

Another solution for identification of MSI targets consists of isolation of regions of interest within the tissue by using laser capture microdissection (LCM). In this set-up, a laser is coupled into a microscope under which the tissue sample can be placed on a normal microscope slide: ROIs can be captured by localized laser ablation after which the tissue parts, that were cut out, are harvested into LCM collection tubes. The obtained tissue samples can then be subjected to full lysis for protein extraction, followed by mass spectrometric analysis to enable identification of MSI targets [33]. Additionally, our research group reported earlier a method to link MALDI MSI with top-down proteomics for a reliable identification of interesting MSI targets with higher-resolution mass spectrometric approaches [20].

With MALDI MSI, molecular characterization of tumors for each patient individually can be achieved, allowing personalized cancer treatment for better clinical outcomes, reduced toxicity and avoiding unnecessary therapy costs. As an example of this, with MALDI MSI on small FFPE tissue biopsies (1 mm diameter), Kriegsmann et al. [34] were able to categorize NSCLC patients into adenocarcinoma and squamous cell carcinoma with high sensitivity and specificity, a classification which is crucial for selection of the type of chemotherapy. A total list of 339 molecular signals were used to create a classification model for adeno- and squamous cell carcinoma and this MALDI classifier could obtain almost 100% diagnostic accuracy. Four peptides were identified by MALDI imaging as strongly differentially expressed between the two diagnoses, one of the peptides was highly expressed in adenocarcinoma biopsies, while a high expression of the three other peptides were seen in squamous cell carcinoma. These four candidate protein biomarkers were validated with immunohistochemical analysis, from which two markers were already well-known discriminative IHC markers for routine NSCLC diagnosis. This confirms the validity of MALDI MSI with the classification model approach for the discovery of new candidate biomarkers [34]. Another example is given by Pallua and co-workers who performed MALDI MSI on prostate cancer tissues and non-malignant benign tissues, resulting in the identification of biliverdin reductase B as a possible biomarker for diagnosis. Overexpression of this biomarker is associated with the diagnosis of malignant prostate cancer [35]. Extensive proteomic research with MALDI MSI was performed by Casadonte et al. [36] on pancreatic ductal adenocarcinoma and pancreatic neuroendocrine tumor tissues, leading to the development of a class prediction model to differentiate between both entities with high accuracy, essential for the appropriate treatment modality. MALDI MSI proteomic analyses has also proven their relevance in triple-negative breast cancer in the search for putative markers for recurrence-free survival: Phillips et al. [37] identified nine proteins, not previously associated with breast cancer, that were significantly associated with worse recurrence-free survival when these proteins were highly expressed. Meding et al. [38] demonstrated that MALDI MSI can be used to classify six different tumor entities in different organ sites with high confidence, based on 117 *m*/*z* species in the average imaging spectra. Furthermore, they could show that MALDI MSI can successfully classify primary (colon and liver) tumors from liver metastatic tumors, which is an important challenge in diagnostics as metastasis has to be correctly assigned even if the primary tumor cannot be found. The correct identification of the tumor origin is an important aspect in the personalized medicine field [38]. Recently, our research group reported the use of MALDI mass spectrometry imaging to measure therapeutic response in NSCLC patients who received immunotherapy [39]. MSI analysis was performed on pre-treatment FFPE biopsies and differential analysis revealed neutrophil defensin 1, neutrophil defensin 2 and neutrophil defensin 3 as predictive biomarkers for the response to immune-checkpoint-based immunotherapy in NSCLC patients. These outcomes were verified with immunohistochemical analyses specific for these neutrophil defensins and after pathological scoring of neutrophil defensin expression on different cell types, statistical analysis exposed a significant association between neutrophil defensin expression and a positive immunotherapy response. This is of major importance for immunotherapy decision, for a better quality of life for the NSCLC patient, to avoid treatment of patients who are not likely to respond and to reduce unnecessary treatment costs [39].

## 4. Limitations for the Full Integration of MALDI MSI in Cancer Research

To sum up, multiple diseased tissues have already been investigated with the use of mass spectrometry (imaging), which allows for the use of multiple molecular markers (compared to only one with immunohistochemical analysis) and thus results in accurate classification of biological samples. Mass spectrometry (imaging) thus holds potential as a tool to improve screening programs. A disadvantage of full integration of tissue proteomics in cancer research is the limit-of-detection; low abundant proteins will not be detected preferentially, as no technique equivalent of polymerase chain reaction for amplification of proteins exists to amplify the protein signal [3]. To detect low abundant proteins, high abundant proteins need to be removed in sample treatment steps, but with the risk of depleting also the low abundant proteins of interest, that may be bound to circulating carrier proteins [8]. Another challenge to fully implement MALDI MSI in cancer research remains the appropriate sample handling and avoiding biological degradation of a large amount of clinical tissue samples. Fresh frozen tissues are better suited for MSI analysis, as less sample preparation steps are required and analysis of intact proteins in their native state can be performed, as well as peptides, lipids and metabolites. One of the disadvantages of fresh frozen tissue sections is that they need to be snap-frozen as fast as possible in order to save sample’s morphology and to minimize (protein) degradation. Repeated freeze-thaw cycles also need to be avoided for minimizing biological degradation. Special attention needs to be given to avoid additional contamination of the sample that may hamper mass spectrometric analysis. The use of optimal cutting temperature (OCT) compounds for instance needs to be avoided, as they can interfere significantly with ionization of peptides and proteins. A consideration is also that fresh frozen tissues are mostly not available in large amounts in biobanks. Furthermore, they need long term storage at −80 °C in expensive freezers with the risk of protein degradation for storage longer than a year [40]. On the other hand, formalin-fixed and paraffin embedded tissues (FFPE) can be stored for decades at room temperature with minimal degradation, leading to the availability of large archives of clinical FFPE biopsies from various diseases in pathological labs and biobanks. The disadvantage is the fact that proteomic analysis of FFPE tissues is hampered by crosslinking processes and intact molecules cannot be detected, with the additional disadvantage that multiple steps of sample preparation (deparaffination and antigen retrieval) are required and increased inter sample variability is usually observed [41,42]. For the full application of MSI in routine histopathology, both standard and more stringent quality control procedures for tissue fixation and handling should be developed within histopathology labs, and standard procedures for MSI sample preparation and processing by interlaboratory validation [43]. The methodology needs to be standardized to allow comparison, reproducibility and reliability of the findings [9].

Another disadvantage of MSI analysis is that it requires invasive approaches for tissue sample collection [6]. Although some challenges remain for the full potential use of MALDI MSI in clinical routine, applications of MALDI MSI into clinical settings are growing, due to its feasibility and molecular visualization on tissue samples of much more molecules in one single analysis, coupled with biocomputational tools to evaluate biostatistical relevant molecular signals [43].

## 5. Future Outlooks

Mass spectrometry analysis enables rapid and sensitive profiling of a high number of molecules simultaneously and in this way, cost-effective high-throughput screening of a high number of clinical tissue samples can be performed, which can identify unique protein patterns. When combined with biocomputational tools, biostatistical evaluation of molecular signals can be performed to discover new protein signatures [43]. The generated proteomic data complements the -omics data produced by other high-throughput technologies with the main purpose to gain crucial insights into tumor biology/pathology. Although proteomics has already revealed new molecular mechanisms and new putative markers in the context of cancer, a combination of different -omics studies in cancer research may better represent the full complexity of cancer and help towards a better understanding of the different molecular alterations that characterizes cancer. In this way, proteomics data can be integrated with genomics, transcriptomics, lipidomics and metabolomics. However, integrating multi -omics data is very challenging, but could lead to the development of more complete predictive models for a better stratification of patients who are most likely to benefit from the administered therapy, contributing further to the field of personalized medicine [44].

MALDI MSI has been used quite intensively in clinical research in the last few years, with the advantage that results can be verified with LC-MS/MS approaches, western blot and/or immunohistochemical studies [18]. With MSI, much more molecules can be detected in one single analysis compared to IHC, which is an advantage as biopsy material can be scarce. Thereby, as personalized medicine in cancer treatment is gathering momentum, MALDI MSI can provide individual molecular characteristics of each individual tumor, necessary to prevent, diagnose, predict therapy outcome or to make a prognosis. Additionally, protein profiles can help to identify new molecular targets to provide a comprehensive understanding of every tumor individual biology, necessary to better understand intratumoral heterogeneity, essential for designing effective therapeutic strategies in the context of personalized medicine.

## 6. Conclusion

To conclude, proteomic profiling may lead to protein biomarker screening, important for cancer diagnosis, cancer subtyping, better (personalized) treatment, but also to the development of novel treatments. The future and ultimate goal of cancer proteomics is to move from bench to bedside applications in cancer management to complement traditional visualization technologies used by pathologists with the aim to guide them in diagnosis or therapy decision in the context of personalized disease management and medicine.

## Figures and Tables

**Figure 1 jpm-10-00054-f001:**
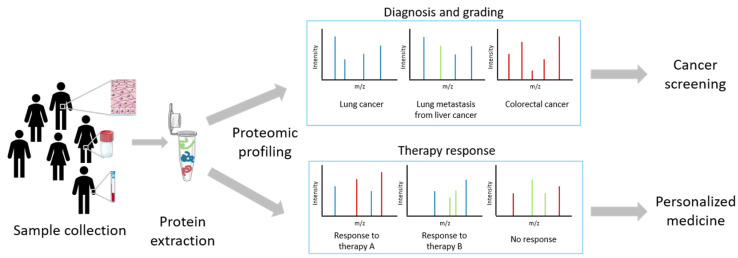
A schematic overview of the implementation of proteomics in cancer screening and personalized medicine. Samples can be biological fluids (f.e. blood and urine) or tissue biopsies from which proteins are extracted to perform proteomic profiling, in this example with mass spectrometry. Different mass spectra can enable diagnosis and grading for cancer screening or assess efficacy and toxicity of therapeutic agents.

**Figure 2 jpm-10-00054-f002:**
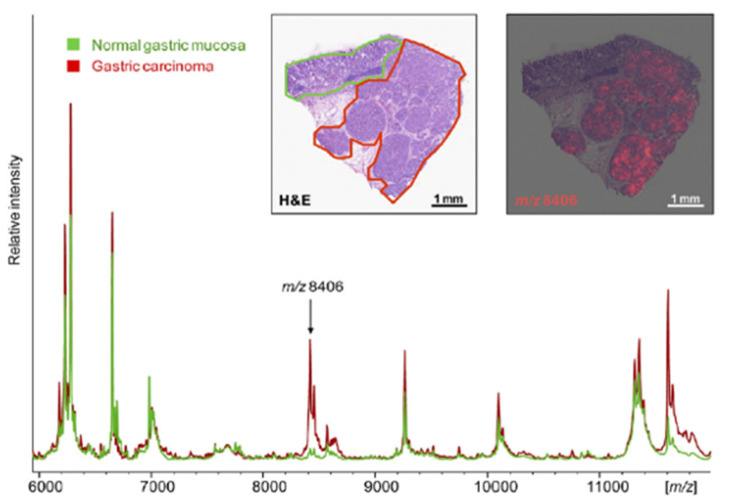
An overview of the use of MALDI mass spectrometry imaging (MSI) in clinical applications. The histological (H&E) staining has revealed the two different regions and MALDI MSI has revealed cell type-specific profiles for gastric carcinoma (red) and normal gastric mucosa (green) from an individual patient’s tissue. The example *m*/*z* 8406 is exclusively present in cancer cells, also indicated on the right inset (red visualization). This protein was also found to correlate significantly with the patient’s overall survival. Adapted with permission from Balluff et al. 2011 (Am. J. Pathol.).

**Figure 3 jpm-10-00054-f003:**
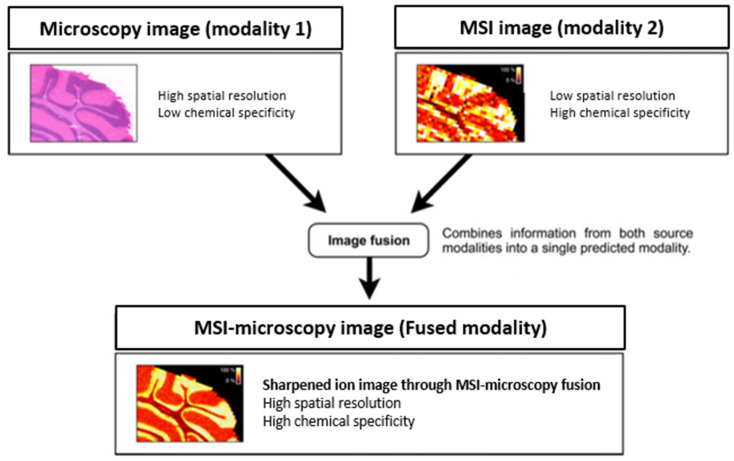
Example of MSI-microscopy fusion by combining high chemical information, obtained with lower spatial resolution (100 µm), with high spatial resolution obtained with microscopy (10 µm). The resulting MSI-microscopy fused image is a predictive modality that delivers both chemical specificity and high spatial resolution in one integrated modality. Adapted with permission from Van de Plas et al. 2015 (Nat Methods).

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
