# Peer review of "Implementation of MALDI Mass Spectrometry Imaging in Cancer Proteomics Research: Applications and Challenges"

_jpm, 2020, doi:10.3390/jpm10020054_

Round 1

Reviewer 1 Report

I really enjoyed reading the review by Eline Berghmans and co-authors, who review the main application and challents towards the implementation of (proteomic) MALDI-MSI in the field of cancer research. Information and literature are relevant. I would suggest adding some more information as mentioned below.

Major points:

I would first like to point out that MALDI-MSI is much more than simply protein analysis, especially in the field of cancer research where lipids (mainly) and metabolites are the center of numerous investigations.
It is the right of the authors to focus on this aspect only, but I would suggest to make it more transparent in the title and include at least the term "protein" / "proteomic(s) to position the review and so that the reader is aware of the focus of the review.
Following this comments, an additional paragraph mentioning the possibility to investigate other class of molecular markers (lipids, metabolites) and perspectives it brings for personalized medicine would be an added value. And to mention extra-challenges associated with correlating the data from these different "omics" modalities would be a plus.

Page 6 line 195 - the authors mention the possibility to analyze hundreds of samples for large clinical studies. Could the authors give (at appropriate place in teh review) some additional information on how challenging it is to obtain clinical samples and to handle them in a appropriate way, so that sample do not degrade and information is maintained? This is a crucial point for a review focused on the "implementation" of the technique for cancer research.

"Also, our research group 231 reported earlier a method to link MALDI MSI with top-down proteomics for a reliable identification 232 of interesting MSI targets with higher-resolution mass spectrometric approaches"==> recent student for microproteomics and targeted extraction of region of interest with laser microdissection would also be worth mentioning here for complementary investigations of potential markers.

Minor comment:

Figure 2: add scale (mm) near the H&E stainings and annotate the regions corresponding to tumours (even with an arrow). Add color scale near the MSI images. The axis of the spectra in C and D and legend in the right in D are very small and difficult to read

Author Response

Dear reviewer,

We would like to thank you for reading this manuscript thoroughly and for the suggestions to improve our manuscript. Please find a summary of the revisions below.

  • We agree with the fact that the title of this review paper can be somewhat misleading, so we added the term ‘Proteomics’ as suggested. Title is now ‘Implementation of MALDI Mass Spectrometry Imaging in Cancer Proteomics Research: Applications and Challenges’.
  • As we are indeed in this manuscript mainly focusing on protein analysis, it is definitely worthwhile to mention that investigation of other classes of molecular markers are possible with MSI. Therefore, as suggested by the reviewer, we added an extra paragraph (lines 180-195) where the importance of MSI for lipids, metabolites and N-glycans is included with an example of each topic.
  • In part ‘5. Future outlooks’ (lines 360-367), we have additionally included the challenging integration of multi -omics in cancer research. Here, we also discussed that integration of lipidomics, metabolomics, genomics and proteomics will lead to more complete predictive models in terms of personalized medicine.
  • We totally agree with the reviewer that obtaining and handling large amounts of clinical samples is challenging and that this is a crucial point to include in this type of review. Therefore, as suggested by the reviewer, we included additional information about this challenge in part ‘4. Limitations for the full integration of MALDI MSI in cancer research’ in lines 323-347.
  • Lastly, as suggested by the reviewer, we also included complementary methods to identify MSI targets: LESA and LCM are now also presented in the manuscript in lines 250-268.
  • As suggested by another reviewer, we have changed Figure 2 to an earlier published figure, so we have not addressed the minor comment suggested by the reviewer according our previous Figure 2.

Thank you again for the fruitful suggestions!

Kind regards,

Eline en co

Reviewer 2 Report

Dear author,

The review article covers very nice the application and challenges of MALDI MSI in cancer research as the title describes.

The article's structure follows a red thread that guides the reader from the presentation of the problem in cancer research, the importance of biomarkers to analyzing them with MALDI MSI and its limitations.

The references in the article are relevant and the figures are well executed.

However, I suggest some minor revision:

  • Reference 8 has only short title (line 333). Because it refers to the cancer proteome project (cancerHPP) (line 67), it needs to refer to a website or similar.
  • Row 197 lacks references to SCiLS software should be referred to as SCiLS Lab (SCiLS, Bremen, Germany), Cardinal should have the reference Bemis, KD, et al., Cardinal: an R package for statistical analysis of mass spectrometry-based imaging experiments . Bioinformatics 2015, 31, 2418-2420, and msIQuant should have the reference Kallback, P., et al., MsIQuant: Quantitation Software for Mass Spectrometry Imaging Enabling Fast Access, Visualization, and Analysis of Large Data Sets. Anal Chem 2016, 88, 4346–4353.
  • Line 199 appears to have missing text and has only “…” in front of reference 24.

Author Response

Dear reviewer,

Thank you very much for reading this review thoroughly and for the small suggestions. We have addressed them all in the manuscript. Please find the summary of the revisions below.

  • Reference 8 (lines 408-409) is changed to ‘Human Proteome Organization (HUPO). Available online: https://www.hupo.org/Human-Cancer-Proteome-Project (accessed on 04 December 2019).’
  • The corresponding references for SCiLS, Cardinal and MsIQuant are added in the manuscript on line 216.
  • The three points were wrongly used to indicate ‘etc.’, which is now correctly demonstrated in the text by writing ‘etc.’ instead of ‘…’ (line 218).

Thank you again!

Kind regards,

Eline en co

Reviewer 3 Report

This review by Berghmans et al. describes the value of proteomics in the pursuit of personalized medicine, and aims to evaluate the (potential) role of MALDI-MSI in the context of personalized medicine. While the topic is relevant, and MALDI-MSI can definitely contribute to personalizing treatment, the review has a number of flaws. In my opinion, fixing these flaws would result in extensive rewriting the majority of the article, and I therefore advise to reject this manuscript.

Some of my issues with the current manuscript are described below.

Comments:

Major comments

  1. While a literature review should cover recent and original research that accurately describes the reviewed research field, this review is largely based on previous reviews (approx. 40% of the references consists of review articles), and therefore based on the interpretation of the original work by another author. Moreover, writing a review in this fashion does not give credit to the authors of the original work.
  2. There are very good examples available in literature that showcase the abilities of proteomics MSI which are not specifically mentioned in this review. However, the previous work by the authors of the review does get specific mentions in the manuscript. While the mentions are not inappropriate or irrelevant, leaving out work by others leaves this review incomprehensive.
  3. The data in Figure 2 appears to be original and previously unpublished. While the example indeed shows the ability of MSI to discriminate between different regions of interest in a tissue section, the spectra seem very light on information. The technique is credited to be able to provide much more information than conventional immunohistochemistry. However, when showing spectra like this that statement is difficult to defend. In literature there are numerous examples that would make this example clear, and would do the capabilities of MSI justice (good examples have been published in the past by the groups of Walch, Heeren, McDonnell and Caprioli).
  4. While the title of the manuscript is “Implementation of MALDI Mass Spectrometry Imaging in Cancer Research: Applications and Challenges”, the review predominantly focuses on the application to (variations) of lung cancer. The title and abstract should better reflect this.

Minor comments

  1. Line 51: proteins/peptides don’t have unique m/z’s (more so when measured by MALDI-TOF-MS). Protein isoforms, and especially proteolytic peptides contain many isobaric and even isomeric species.
  2. Line 76: this statement appears to refer to previous work which is not referenced.
  3. Line 86: drawing blood from a patient is considered (mildly) invasive practice.
  4. Line 199: “…” is the sentence incomplete?
  5. Throughout the manuscript m/z is both written in italic and in normal type. Italic is the consensus throughout literature.

Author Response

Dear reviewer,

We would like to thank you for reading this manuscript thoroughly and for the suggestions to improve our manuscript. Please find a summary of the revisions below.

  • In introduction and challenges we indeed referred to a number of review papers. This was intentional, as we aimed to keep the introduction to the broader field of proteomics and cancer research brief and still wanted to refer the readers to a number of excellent reviews on the topic, since covering a broader field is not the intention of this review, we felt this would be appropriate. However, we do agree with the reviewer that proper reference to the original research is desirable. In the revised manuscript we added citation to the original manuscript when a proteomic example is given. Additional information added in the manuscript is also based on the original work of the authors.
  • There are indeed multiple good examples of proteomics MSI in literature. We included references to a number of these examples we felt are relevant, although other examples may exist. By no means do we intentionally left out relevant examples nor did we intentionally over review our own work as the reviewer seems to imply. Rather, we only included previous work if we felt it was relevant (as seems to be acknowledged by the reviewer). Evidently, we would have gratefully accepted additional references that the reviewer felt were missing to our review. In response, taking the reviewer’s remarks into consideration not to put too much emphasis on lung cancer, we added a number of references to other types of cancer studied using MSI, including an example of prostate cancer, pancreatic cancer and triple-negative breast cancer. We therefore have added three additional examples of proteomics MSI by Casadonte et al., Pallua et al. and Phillips et al. included at lines 285-296.
  • By adding the three extra proteomic MSI examples on different types of cancer, we have chosen to not include ‘lung cancer’ in the title. For the title, we added the term ‘proteomics’ to make clear that the review covers proteomic researches.
  • Figure 2 is indeed previous unpublished and therefore it may, as suggested, be better replaced by an example from literature. In addition, it indeed provides the opportunity to select a tissue that shows more features in MSI than the example currently chosen. We selected a figure from the research group of Walch (Balluff et al.) from gastric normal mucosa and carcinoma.
  • Minor comments:
    • Line 52: We have left out the sentence about unique m/z’s, as this sentence is indeed confusing.
    • Line 76: To make it more clear that is part of the definition of ‘biomarkers’ and that is referenced by reference [9], we have changed ‘These’ to ‘Such’ and added the reference again after this sentence.
    • Lines 87-88: We agree with the reviewer that drawing blood from a patient is considered mildly invasive and stated this clearly in the revised manuscript at lines 87-88.
    • Line 218: The three points were wrongly used to indicate ‘etc.’, which is now correctly demonstrated in the text by writing ‘etc.’ instead of ‘…’.
    • We have changed all m/z listed in the manuscript to italic to reach the consensus.

Thank you again for the fruitful suggestions!

Kind regards,

Eline and co

Round 2

Reviewer 3 Report

The authors made the requested changes to the manuscript, and have addressed the issues that were raised earlier. 

The only comment I would still have are to:

  • Include a space between RCC and the reference on Line 112
  • provide references to research papers describing MALDI-MSI of the different analyte classes (Line 186). 

Author Response

Dear reviewer

Thank you for reviewing again and for the minor suggestions. We addressed them both:

  • We included the space between 'RCC' and the following reference at line 110.
  • We referred to the research papers performed to study the different mentioned analytes with MSI at line 183.

Thank you again for reading thoroughly and the suggestions to improve the manuscript!

Kind regards

Eline and co